# Factors Associated with Post-Traumatic Growth in Healthcare Professionals: A Systematic Review of the Literature

**DOI:** 10.3390/healthcare10122524

**Published:** 2022-12-13

**Authors:** Róisín O’Donovan, Jolanta Burke

**Affiliations:** Centre for Positive Psychology and Health, RCSI University of Medicine and Health Sciences, D02 YN77 Dublin, Ireland

**Keywords:** post-traumatic growth, healthcare, systematic review

## Abstract

Post-traumatic growth (PTG) research is flourishing across various disciplines; however, it is only emerging in healthcare. Recently, a flurry of studies assessed PTG among healthcare professionals. However, to date, no systematic review has identified the factors that predict their experiences of PTG among healthcare professionals. The current paper aims to address this gap. Of 126 papers, 27 were selected for this systematic review. The analysis identified a range of demographic, individual (work-related and personal), interpersonal and environmental factors that contributed to healthcare professionals’ experiences of PTG. Analysis of findings offers a novel perspective on individual factors by dividing them into personal and work-related factors. Results also highlight a variety of psychological interventions that can be used in healthcare to cultivate PTG. In addition, the gaps in current research, implications for further research, policy and practice that can facilitate the experience of PTG among healthcare professionals are discussed.

## 1. Introduction

The COVID-19 pandemic resulted in many healthcare professionals, especially those working at the frontline, experiencing some degree of trauma [1]. Furthermore, a considerable proportion of healthcare professionals reported traumatic stress and symptoms of post-traumatic stress disorder [2,3]. This is not surprising given that healthcare professionals are indirectly but repeatedly exposed to traumatic events [4]. This points to the potential of the healthcare sector needing to deal with the significant, long-term impact of the pandemic on healthcare professionals. Dealing with such impacts will require targeted interventions that are based on an in-depth understanding of the factors associated with post-traumatic growth (PTG) among healthcare professionals.

The outcome of trauma ranges from experiencing pathologies, such as post-traumatic stress, to growth and positive emotions [5]. This PTG can help develop a new meaning of life, spiritual beliefs, changes in worldview and realization of personal values [6,7]. PTG refers to the psychological growth reported by individuals who have experienced a traumatic event. It is fueled by humans’ innate need to sustain and improve their wellbeing [8]. Psychological growth occurs due to a struggle following adversity that challenges pre-trauma schemas and worldviews by shattering beliefs, goals or assumptions about life [9]. This results in psychological growth that exceeds individuals’ previous psychological functioning or adaptation [10], which becomes a positive outcome of an adverse situation. Understanding the specific modifiable factors associated with PTG can help design interventions that help practitioners cope more effectively with trauma [11]. The current review aimed to address this need.

The conceptualizations of PTG range from experiencing stress-related growth [12], benefit-finding [13] and changes in outlook [14]. The most salient approach to perceiving growth post-trauma centers on five domains, which are (1) relating to others, (2) new possibilities, (3) personal strength, (4) spiritual change and (5) appreciation of life (1). “Relating to others” means that individuals who experience adversity develop a greater sense of closeness to specific people in their lives and begin to appreciate their family and friends more. Furthermore, they may also start to see new opportunities unfolding before them, whereby they may change a direction in their lives, such as pursuing education, a new job, or establishing another new path. For some, growth comes as realizing their “personal strength”; therefore, they start to believe that they are mentally or physically more robust than they had ever thought or perceive themselves as more equipped to handle challenges. For others, their growth is associated with having a stronger religious belief. Finally, many people who experience post-traumatic growth talk about their new-found “appreciation of life” that prompts significant changes in the way they prioritize what is essential in life. This five-domain model is the focus of the current systematic review.

Tedeschi and Calhoun’s model [9] perceived PTG as an outcome of a direct trauma experienced by individuals. However, the PTG Inventory has subsequently been used to assess secondary (SPTG) or vicarious post-traumatic growth (VPTG), denoting positive changes experienced following observing other people experiencing trauma, which is often the case with healthcare professionals [15]. Nonetheless, research measuring these concepts does not differentiate between indirect and direct traumas, nor does it distinguish between specific factors related to indirect and secondary trauma. This makes its validity and reliability questionable. This is why the current review considered PTG in the original context, as developed by Tedeschi and Calhoun [9] and did not include SPTG or VPTG.

A previously conducted meta-analysis of 26 studies demonstrated that between 10–77.3% of participants experienced PTG [16]. The meta-analysis conducted by Wu and colleagues [16] represents the levels of PTG in the general population and the variation in experiences of PTG is likely due to the wide range of trauma experienced by those who participated in the included studies. Professionals aged under 60 years reported the highest levels of PTG. Furthermore, 78% of participants engaged in preemptive benefit-finding when awaiting a diagnosis [17]. PTG is a normative process that occurs in individuals in the face or the aftermath of trauma [18]. The symptoms of PTG can either coexist with post-traumatic stress disorder (PTSD), showing a small correlation with each other, or replace it [19,20,21]. Similarly, a meta-analysis of PTG identified that it is not associated with depression or anxiety [22]. Instead, it may coincide with depression and anxiety or exist in place of it. Thus, PTG is part of a complex system of outcomes related to trauma.

The benefits of experiencing PTG are vast. They range from making sense of loss [23], the development of wisdom [24] through to enhanced purpose and meaningfulness of traumatic experience, which continue to last a decade post-trauma [25]. Longitudinal studies showed that experiencing PTG can also reduce distress a year later and the occurrence of PTSD three years later [26]. In historical research, finding benefits shortly after adversity (heart attack) reduced its reoccurrence eight years later [27]. Therefore, assessing and promoting PTG among healthcare professionals can benefit their health.

Nonetheless, one of the criticisms of it is the potentially dysfunctional nature of PTG [28], whereby, on the one hand, PTG can help people adjust to changed circumstances (constructive component), on the other hand, it may also sustain their avoidance and denial (illusionary component). For example, in a longitudinal study with cancer patients, researchers found that those who reported illusionary PTG were engaged in more maladaptive coping strategies and experienced higher levels of depression and anxiety in a ten-year follow-up compared to participants in a constructive PTG group [29,30]. PTG was a coping strategy rather than a sign of authentic positive changes for them. Therefore, the intricacies of PTG must be recognized when assessing it in the context of healthcare professionals.

The factors associated with PTG range from intrapersonal to interpersonal and environmental. The relationship between PTG and personality characteristics is inconsistent [31]. Nonetheless, the predisposition to perceiving benefits in adverse situations or optimism may contribute to PTG [32]. PTG occurs when adverse events are central to individuals’ self-identity [33]. Of the 24-character strengths (e.g., gratitude, love, optimism), the strength of hope best predicted PTG [34]. Furthermore, seeking social support, resilience, self-efficacy and adaptive coping strategies are associated with experiences of PTG [31,35]. Among young people, PTG was predicted by parenting rather than their intrapersonal assets [36], suggesting the role that the environment plays in helping individuals experience of PTG.

Three main types of intervention approach for enhancing PTG are (1) self-expression or disclosure—written or spoken, (2) cognitive behavioral therapy and (3) novel psychological therapies aimed at facilitating PTG [37]. These psychosocial interventions effectively promote moderate changes in PTG [37,38]. Furthermore, a meta-analysis of mindfulness interventions (median length of 8 weeks) showed a small positive impact on PTG [39]. However, Joseph [40] warns against setting up a goal for experiencing PTG. Instead, PTG should be seen as an outcome of adjusting to trauma.

The current literature review focused on identifying factors that facilitate PTG among healthcare professionals. In addition, it compares findings to research conducted with a general, clinical and non-clinical population, intending to identify the patterns emerging in the literature relating to healthcare staff and how they compare with other populations.

## 2. Materials and Methods

Enablers of PTG were identified through a systematic review of the literature. This process was used to synthesize the evidence from available studies to explore ways in which PTG can be fostered among healthcare professionals [41,42]. The Cochrane and Preferred Reporting Items for Systematic Reviews and Meta-Analyses (PRISMA) guidelines [42,43,44] were followed. The protocol for this review has been published on Prospero (registration number: CRD42021288062).

### 2.1. Inclusion and Exclusion Criteria

Included studies examined enablers of PTG among healthcare professionals. They needed to report on the association, assessment or evaluation of factors that enable PTG. They also needed to be peer-reviewed, from any country, written in English and published between 1998 and 2021. Studies that did not meet all of these criteria were excluded. 

### 2.2. Search Strategy

The following search strategy was used to search each database. “healthcare professional*” OR “healthcare worker*” OR “healthcare provid*” or “physician” or “nurs*” or “doctor” AND “post-traumatic growth” OR “post-traumatic growth”. A full search strategy has been included in Appendix A.

### 2.3. Information Sources

Electronic databases (Medline, PsycINFO and CINAHL) were searched between September 2021 and October 2021.

A supplementary grey literature search was conducted using a grey literature database (OCLC WorldCAT). This database has a broad scope and the ability to conduct specific searches [45,46]. The reference lists of included studies were searched by the authors.

### 2.4. Study Screening

Rayyan, an online database for conducting systematic reviews, was used to screen all studies. Two reviewers screened titles and abstracts based on the eligibility criteria and then both reviewers independently reviewed each text. Both reviewers had expertise in Psychology and Health Services research. Meetings were held to discuss and resolve any conflicts in reviewers’ assessments. There was an option to involve a third reviewer if an agreement could not be reached. However, this was not necessary as, following discussion, the original two reviewers reached an agreement on all papers for inclusion. The screening stage of this review is presented in a PRIMSA flow diagram (see Figure 1).

### 2.5. Data Extraction Process

A data extraction template was developed to capture the relevant information from included studies. This template was based on the third version of guidelines produced by Cochrane in 2014 and recommendations from Hoffmann et al. [47]. Information was collected for aims, design, theoretical underpinnings, participant information and enablers identified. The final template can be seen in Appendix B.

### 2.6. Quality Assessment

The Mixed Methods Appraisal Tool (MMAT) [48] was used to assess the quality of included studies. The MMAT was chosen because it facilitates the consistent appraisal of the types of study methodologies and designs included in this review (Quantitative Descriptive and Mixed Methods).

### 2.7. Study Synthesis

A narrative approach to synthesis was used to describe the findings from the included [49]. The heterogeneous nature of the studies identified meant that a narrative synthesis was the most appropriate approach to exploring similarities and differences between study designs [50]. Taking a narrative approach also allows for the generation of key themes related to findings. Specifically, the narrative synthesis followed three iterative steps: organizing studies into logical categories by becoming familiar with them; comparing them to one another and synthesizing their findings; analyzing the findings within each category by exploring relationships within and between the studies and synthesizing data under the relevant themes [49].

### 2.8. Publication and Researcher Bias

To minimize the risk of publication bias, searches were conducted on academic and grey literature databases. Researcher bias was limited by having two reviewers independently assess the eligibility of the included papers and conducting a quality assessment of all included papers.

## 3. Results

An overview of the 27 studies included in this review can be found in Table 1. The majority of studies included a nursing population (n = 24) and used a cross-sectional survey design (n = 19). Three studies used a mixed-methods approach, three were longitudinal and two were intervention studies. Other studies included populations of physicians, general practitioners, social workers, medical technicians, medical researchers, administrators, occupational therapists, psychologists, physiotherapists and pharmacists (n = 10). Six studies met all quality criteria and all other studies met at least five criteria on the Mixed Methods Appraisal Tool (2). The results of these assessments are presented in Appendix C. The enablers of PTG identified in these studies are described below and are presented in Figure 2.

### 3.1. Individual Level Factors

#### 3.1.1. Work-Related

*Self-Confidence* was identified as a work-related individual factor influencing PTG. In the study conducted by Cui et al. [55], nurses who were confident about their frontline work had higher PTG. 

The next work-related individual factor was *Self-Efficacy.* Zhang et al. [78] found that self-efficacy could positively predict the level of PTG among nurses. The self-efficacy measured in this study is closely related to findings related to self-confidence as it refers to individuals’ perception of their capability to successfully implement a particular behavior or goal or to overcome a disadvantage.

Two studies found that *Personal Accomplishment*, which refers to levels of self-achievement and competence at work [79], facilitated PTG [5,75]. Chen et al. [5] highlighted that feeling more personal accomplishment had the highest explanatory power for PTG.

**Figure 2 healthcare-10-02524-f002:**
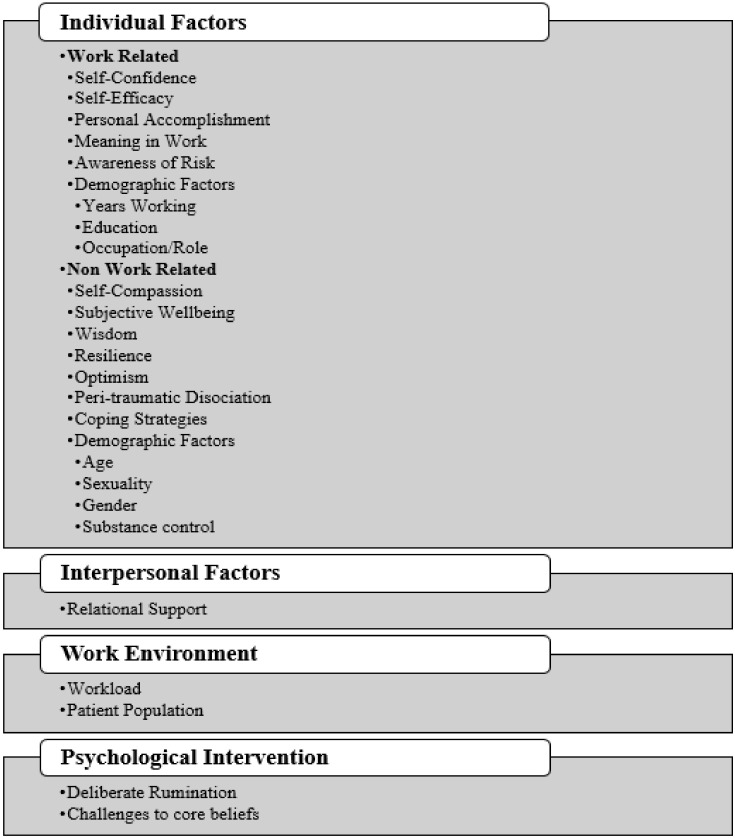
Categories of Identified PTG Enablers.

*Meaning in Work* played a moderating role among pediatric nurses. Meaning in work is defined as a subjective experience that work is intrinsically significant and worth doing; that one can achieve a sense of autonomy and self-expression through work; and that the work serves a greater or broader purpose [80]. When participants had high levels of meaning in work, secondary traumatic stress (experienced when helping or wanting to help a traumatized or suffering person) had a positive mediating effect on PTG [58]. 

Studies found that *Awareness of Risk* impacted PTG. Nurses who were aware of the risk associated with the COVID-19 pandemic had higher PTG [55]. Cui et al. [55] propose that participants who were aware of the risks associated with the pandemic would be better able to respond to the emergency. Similarly, Moreno-Jiménez et al. [68] found that fear of contagion positively predicted PTG. Nurses who reported higher event-specific distress also reported higher PTG [61].

Behavioral disengagement occurs when an individual expects a negative outcome and thus reduces or eliminates efforts to handle the stress [81]. Lower levels of behavioral disengagement are associated with greater PTG scores, and this supports the idea that the development of PTG is an active process that requires attention to the stressful experience (i.e., risks associated with COVID-19) and subsequent deliberate action.

A variety of *Demographic Factors* impacted PTG. Firstly, Cui et al. [55] found that *Years Working* impacted nurses’ PTG, with those who had more than 10 working years of working experience having higher PTG scores. They proposed that this may be because they have more experience and higher levels of critical thinking. Participants with more working years were married and had children. Similarly, Li et al. [38] found that nurses with a senior professional title had higher PTG scores and proposed that this was due to having more work experience. *Education* also influenced PTG, with two studies finding that having a postgraduate degree contributed to higher PTG [69,70]. *Occupation/Role* was another demographic factor that impacted PTG. Being a nurse was associated with higher PTG, according to three studies [38,57,64]. Hamama-Raz et al. [57] found that being a nurse was associated with a higher PTG total score on all subscales except for relating to others. Lev-Wiesel et al. [64] also found that nurses had higher PTG compared with social workers. Li et al. [38] found that total PTGI and three domains of PTGI, new possibilities, personal strength and spiritual change were higher in nurses than in general practitioners.

#### 3.1.2. Non-Work Related

Firstly, two studies identified *Self-Compassion* as an enabler of post-traumatic growth (PTG) among nurses [51,53]. Both studies draw on Neff [82] who defines self-compassion as being generous or kind to oneself, perceiving one’s experience of suffering as part of the larger human experience and being mindful to not over-identify with painful thoughts or emotions. Chang et al. [53] explain that nurses with positive self-compassion can view their suffering as a part of common life experiences and that this helps them gain the insights needed to reinterpret the traumatic event positively, thus supporting the development of PTG.

Aggar et al. [51] found that greater *Subjective Well-being* was associated with post-traumatic growth. Similarly, life satisfaction and dedication to their role contributed to PTG among mental health nurses [60].

In contrast, nurses who reported physical discomfort during the pandemic (including insomnia, grey hair/hair loss, weight loss, loss of appetite, irregular menstruation, Lumbar muscle strain/muscle soreness, coughing/sputum and skin eczema) had higher PTG score [71].

*Wisdom* was identified as an enabler of PTG in two studies [53,72]. Wisdom allows people to deeply understand the contradictions of life, and existential problems while maintaining a broad perspective and this helps them develop PTG when they confront adversities or life crises [83]. 

*Resilience* was identified as an enabler of PTG in four studies [59,62,65,66]. Three of these studies measured resilience using the Connor–Davidson Resilience Scale (K-CD-RISC), which captures resilience in relation to hardiness, persistence, optimism, support and spiritual attributes [59,62,65]. One used the Ego-Resilience Scale (ERS) to capture individuals’ levels of resilience [66]. Jung and Park [62] found that resilience was the most influential predictor of PTG among nurses. While Lyu et al. [66] found that resilience had a positive association with PTG at Time 1 and 2, this relationship was not found at Time 3. This was explained by the fact that individuals at Time 3 were experiencing lower levels of stress because COVID-19 was more under control in China at that time. They, therefore, propose that resilience is particularly important to cultivate PTG during times of high stress. Hyun et al. [59] proposed that one explanation for the positive relationships between resilience and PTG is that healthcare professionals with high levels of resilience are more likely to view secondary trauma as a challenge to overcome and to better manage the stress associated with traumatic events.

Three studies identified *Optimism* as an enabler of PTG [66,73,76]. Lyu et al. [66] attributed increased PTG at Time 2 to increased participant optimism. However, this optimism faded as it became clear that the pandemic situation was not under control and this resulted in a decline in PTG at Time 3. Shiri, Wexler and Kreitler [73] found that having beliefs rooted in optimism increased PTG. Specifically, participants who had beliefs about their goals that reflected an optimistic view of their future had higher PTG. Xu et al. [76] also found that cultivating optimism supports PTG.

*Peri-traumatic Dissociation*, a sense of emotional detachment that can be activated during or immediately after a traumatic event, contributed to PTG among social workers. The authors proposed that a certain level of emotional detachment is necessary for coping with adverse situations. However, higher functional levels of dissociation such as emotional detachment do not have the same effect and may lead to symptoms of Post-Traumatic Stress (PTS) [64].

Both problem-focused and emotion-focused *Coping Strategies* were positively associated with PTG among nurses [58]. Problem-focused coping strategies are used when individuals deal with problems directly, such as by seeking information, trying to get help and inhibiting or taking direct actions. Emotion-focused coping involves behavioral and cognitive attempts to decrease or manage emotional distress, such as avoiding them, seeking emotional support, or attempting to see the humor in the situation.

Non-work-related *Demographic Factors* were also identified. Three articles found that *Age* influenced PTG, with older individuals experiencing higher PTG [53,63,69]. Specifically, Lee and Kim [63] found that older caregivers with a religious affiliation experienced greater PTG after patient death and Okoli and Seng [70] found that compared to those in the 18–25-year category, those aged between 36–50 had higher personal strength scores. One study identified *Sexuality* as a factor impacting PTG and reported that being non-heterosexual was associated with higher PTG [69]. Three studies identified *Gender* as an enabler of PTG [38,57,69]. Being female was associated with higher scores in the personal strength subcategory [69]. Being a woman was associated with higher PTG in all subscales except for spiritual change, which showed no significant effect [57]. In contrast, Li et al. [38] found lower PTGI scores in female nurses compared to male nurses. Lastly, *Substance Control* impacted PTG with those with lower alcohol consumption experiencing more PTG [69,70].

### 3.2. Interpersonal Factors

*Relational Support* was identified as an enabler of PTG [78]. According to Hyun, Kim and Lee [59], relational support for coping fosters PTG. Interestingly, the main source of relational support was from other nurses and not their friends and family members. This may have been due to their separation from their personal relationships during the MERS outbreak. Similarly, Jung and Park [62] found that having a positive, supportive relationship with the head nurse had a positive effect on participants’ PTG. In terms of personal relationships, marriage status and having children both predicted PTG [38,69,71]. Peng et al. [71] also found that getting support from family and friends supported PTG. Having relational support also presents opportunities to share experiences, which was identified as a way of supporting PTG [72].

### 3.3. Work Environment Factors

Jung and Park [62] found that the nursing work environment affects the degree of PTG experienced by nurses.

#### 3.3.1. Workload

PTG was higher when the workload was high, and this was especially the case in those with a high lack of staff or Personal Protective Equipment (PPE) [68].

#### 3.3.2. Patient Population

Caring for patients with COVID-19 was associated with higher PTG [5]. In addition, those working in pediatric care and non-direct care experienced more PTG [69,70].

#### 3.3.3. Occupation/Role

Being a nurse was associated with higher PTG, according to three studies [38,57,64]. Hamama-Raz et al. [57] found that being a nurse was associated with a higher PTG total scale on all subscales except for relating to others. Lev-Wiesel et al. [64] also found that nurses had higher PTG compared with social workers. Li et al. [38] found that total PTGI and three domains of PTGI, new possibilities, personal strength and spiritual change were higher in nurses than in general practitioners.

### 3.4. Psychological Intervention or Training

Identified studies found that psychological intervention or training can support PTG. According to Cui et al. [55], receiving psychological intervention or training before or during the frontline work period can help nurses feel a sense of professional responsibility, generate positive psychological experience and help them achieve PTG. Li et al. [38] found that having ways to cope with stress was a predictor of PTG in nurses. In their study, participants using the WeChat network, psychological counselling and phone application for self-relaxation to cope with stress had higher PTG scores than others. Similarly, healthcare professionals who had treatment for trauma also had higher PTG [69]. Xu et al. [76] conducted an intervention which included modules in emotional management and opportunities to share experiences and growth with other participants. This intervention showed improved PTG. An intervention conducted by Yilmaz et al. [77] also improved PTG. This intervention consisted of the therapeutic use of exercise, baksi dance and mandala painting techniques. They also conducted group counselling sessions and sent daily motivational text messages to participating nurses.

Other studies identified psychological processes that can be used to cultivate PTG.

#### 3.4.1. Deliberate Rumination

Two studies identified deliberate rumination as an enabler of PTG. Over time, emotional distress caused by intrusive rumination after trauma is alleviated by a constructive cognitive process that leads to deliberate rumination [52]. This deliberate rumination supports PTG [53]. Nurses who engaged in deliberate rumination by thinking about what they had learned from their experiences during the COVID-19 pandemic had higher PTG [55].

#### 3.4.2. Challenges to Core Beliefs

Challenges to one’s core beliefs play an important role in the development of PTG. This relationship appears to be the case even when taking into account other factors such as perceived social support and coping styles [61]. In the study conducted by Lee and Kim [63], psychological suffering in terms of expanding self-consciousness, change in values and spiritual sublimation had a positive correlation with PTG.

## 4. Discussion

This is the first systematic review of PTG experienced among healthcare staff. It provides a unique perspective on factors associated with PTG, which can help practitioners facilitate occurrences of PTG in the aftermath of trauma. Consistent with previous reviews, it provides a list of individual and interpersonal enablers of PTG, which are consistent with research conducted with other populations [31]. However, it offers a novel perspective on individual factors by dividing them into personal and work-related factors. Furthermore, this review identifies unique environmental and demographic factors that supported healthcare staff in facilitating PTG. According to Goldberg and colleagues (2019), modifiable factors can inform future PTG interventions. As such, this expanded knowledge of the work-related individual, interpersonal and environmental factors can become a starting point for further research and the creation of targeted PTG interventions specific to healthcare professionals.

The psychological interventions included in this review identified a wide variety of interventions used in healthcare to cultivate PTG. However, the lack of consistency between these interventions limits our understanding of their effectiveness and generalizability. Components of specific interventions align with the enablers identified in this review, e.g., emotional management strategies correspond to emotional coping styles and sharing experiences and growth with others align with social support [76]. However, many other interventions were ad hoc. This highlights an urgent need to develop interventions firmly grounded in the evidence for factors associated with PTG. Further research is required to explore PTG interventions for healthcare professionals that are in alignment with the factors identified in this review and to ensure that these interventions are implemented into healthcare policy and practice.

The factors identified in this review can help to inform the development of future interventions to improve PTG. Chang et al. [53] highlighted the need to develop short self-compassion programs tailored to the fast-paced, high-intensity environment in which ICU nurses work. They also suggested that deliberate rumination-promoting programs, including aspects of self-disclosure and co-worker support, be provided for ICU nurses [53,55]. Cui et al. [55] also suggested guiding nurses to explore and reflect on the positive significance of their experience in order to facilitate PTG. Jung and Park [62] call for programs that can improve resilience among healthcare professionals. In line with this, a recent review has integrated the available management strategies to increase resilience in healthcare workers [84]. They highlighted the role of individual-level strategies such as exercise, good sleep hygiene and mindfulness. However, given that healthcare professionals, such as physicians, have been found to have high levels of resilience, it is vital that we also consider the system-level issues that impact their wellbeing [85]. This involves recognizing the role of organizational strategies, such as cultivating a workplace culture that ensures fairness, respect and social justice, in building resilience at the organizational level [84]. This needs to be considered in the future when developing and implementing PTG interventions for healthcare professionals.

The current review has highlighted the need for PTG interventions to draw on both individual and organizational factors to support the psychological wellbeing of healthcare professionals. It is essential to consider the broader, contextual factors that influence PTG as this can play a valuable role in creating a workplace supportive of PTG. Nevertheless, only four studies examined the environmental factors for experiencing PTG among healthcare staff [5,68,69,70]. Recent calls have been made to examine the psychological impact of COVID-19 at an institutional level and to adopt a systems lens that recognizes the influence institutions have on the individuals who belong to them and, therefore, interventions can have a broader impact by moving beyond the individual-level to create changes within institutions [86,87]. Therefore, it is important to consider the influence of the culture in healthcare organizations on healthcare professionals’ experiences of PTG. The culture in many healthcare organizations contributes to the problems facing the healthcare system, including burnout [88]. By not providing support systems and networks for healthcare professionals who experience trauma, healthcare organizations create a culture that normalizes trauma. This further perpetuates the lack of support for those who experience trauma and limits their ability to move from surviving post trauma to thriving. Future research should focus on adjusting healthcare institutions’ environmental factors and facilitating more relational support within healthcare teams. For example, Schwartz Rounds can provide a platform for staff to express their emotional and social needs at work, improve teamwork and reduce isolation within healthcare teams [89,90]. In addition, Psychological First Aid is an intervention that can be used in the aftermath of a disaster and can address the fundamental social support needs of healthcare teams at that time [91,92].

The intervention studies included in this review were conducted with single groups [76,77]. Therefore, experimental studies that include control groups are required to assess the effectiveness of these interventions rigorously. Future research should also use single interventions that target PTG in order to establish an apparent causal effect between specific interventions and PTG. Additionally, only three studies included a longitudinal follow-up [61,66,68]. Further longitudinal research is required to make clear conclusions regarding causality, to understand changes in healthcare professionals’ levels of PTG over time and to identify the interaction this has with enablers of PTG.

This review included only three studies with a qualitative component [59,72,74]. This lack of in-depth qualitative research limits our understanding of the relationship between PTG and the enablers that have been identified thus far. Further qualitative research is required to gain a more in-depth understanding of the connection between PTG and these enablers and to explore other enablers of PTG that have not been captured within the mainly quantitative studies reported in this review. Qualitative studies can identify and understand aspects of experiences that quantitative methods and positivist perspectives may miss [93]. By using a qualitative approach, future studies will be able to gain an in-depth understanding of lived experiences of PTG among healthcare professionals, which will offer insights into the nuances and variations in healthcare professionals’ experiences. Additionally, qualitative studies will improve our understanding of the complex influence previously identified enablers have on PTG by accounting for the nuances of individuals’ experiences. Lastly, the future use of qualitative methods to improve our understanding of PTG will ultimately inform the development of interventions to improve PTG and ensure that they are grounded in healthcare professionals’ experiences.

This review identified preliminary evidence about the emergence of the most vulnerable groups among healthcare professionals, who are less likely to experience PTG in the face of trauma. They included junior nurses and other less experienced members of the healthcare profession, who were educated to less than postgraduate level and displayed less confidence and belief in their abilities. The vulnerability of inexperienced healthcare professionals is consistent with previous research [94]. However, this study provides a unique contribution by identifying the vulnerable healthcare groups in the context of PTG and identifies factors that may support these groups in developing PTG. Future research should explore this topic further.

## 5. Conclusions

The current systematic review provides a unique perspective on the literature relating to the experiences of PTG among healthcare professionals. It highlights the need for future research to develop targeted staff interventions that incorporate individual, interpersonal and environmental factors. Furthermore, it identifies vulnerable healthcare professional groups that require additional support in helping them experience PTG. Finally, it highlights gaps in the current research and offers suggestions for future research relating to PTG among healthcare professionals. 

## Figures and Tables

**Figure 1 healthcare-10-02524-f001:**
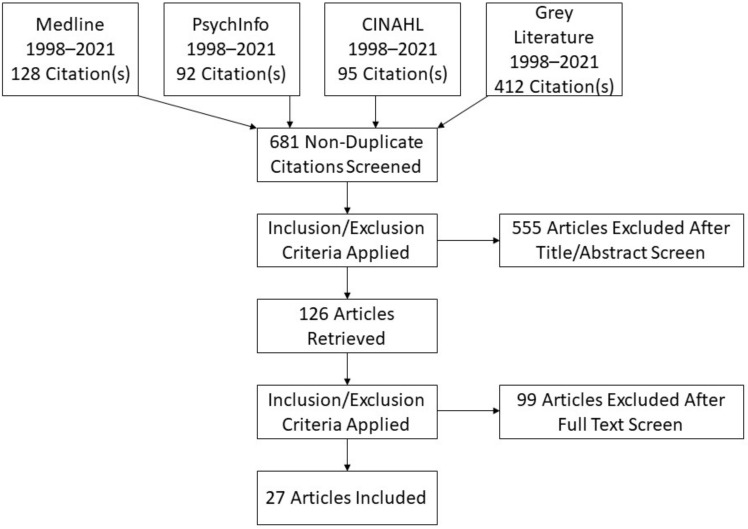
PRISMA Flow Diagram.

**Table 1 healthcare-10-02524-t001:** Overview of Included Studies.

Reference	Model or Measure of PTG Used	Population	Enabler(s) Identified
Aggar et al. [51]	Post-traumatic Growth Inventory–short form [52]	Nurses (n = 767)	Greater subjective well-being; Greater self-compassion
Chang et al. [53]	Post-traumatic Diagnostic Scale (PDS-K) [54]	Nurses (n = 156)	Self-compassion; Wisdom; Age (older); Deliberate rumination
Chen et al. [5]	Post-traumatic Growth Inventory-Short Form [9]	Nurses (n = 12,596)	Caring for patients with COVID-19; Personal Accomplishment
Cui et al. [55]	Post-traumatic Growth Inventory [9]	Nurses (n = 167)	Deliberate rumination; Years working; Self-confidence; Awareness of risk; Receiving psychological intervention or training
Hamama-Raz et al. [56]	Post-traumatic Growth Inventory [9]	Nurses (n = 138)	Secondary traumatic stress; Meaning in work
Hamama-Raz et al. [57]	Post-traumatic Growth Inventory [9]	Nurses (n = 128) and Physicians (n = 78)	Gender (female); Being a nurse
Hamama-Raz and Minerbi [58]	Post-traumatic Growth Inventory [9]	Nurses (153)	Problem-focused and emotion-focused coping strategies
Hyun, Kim and Lee [59]	Post-traumatic Growth Inventory [9]	Nurses (n = 78 surveys; n = 7 interviews)	Resilience; Relational support for coping
Itzhaki et al. [60]	Post-traumatic Growth Inventory–short form [52]	Mental Health Nurses (n = 118)	Life satisfaction; Dedication to role
Jesse [61]	Post-traumatic Growth Inventory [9]	Nurses (n = 49)	Higher event-specific distress; Challenges to one’s core beliefs; lower levels of behavioral disengagement
Jung and Park [62]	Post-traumatic Growth Inventory [9]	Nurses (n = 27)	Resilience; Nursing work environment; Relationship with the head nurse
Lee and Kim [63]	Post-traumatic Growth Inventory [9]	Nurses, Nursing Assistants and Social Workers (n = 254)	Age (Older) and having a religious affiliation; Higher psychological suffering related to the change in values and spiritual sublimation
Lev-Wiesel et al. [64]	Post-traumatic Growth Inventory [9]	Nurses and Social Workers (n = 204)	Being a nurse; Peri-traumatic dissociation
Li et al. [38]	Post-traumatic Growth Inventory [9]	Nurses (n = 455) and General Practitioners (GPs) (n = 424)	Being a nurse; Gender (male); Marriage status (married); Having children; Higher professional title; Having strategies to cope with stress.
Liu, Ju and Liu [65]	Post-traumatic Growth Inventory [9]	Nurses (n = 200)	Resilience
Lyu et al. [66]	Wang and colleagues [67] modified version of Post-traumatic Growth Inventory [9]	Doctors, Nurses, Medical Technicians, Medical Researchers and Administrators (Study 1 n = 134; Study 2 n = 401)	Resilience; Optimism
Moreno-Jiménez et al. [68]	Post-traumatic Growth Inventory-Short Form [52]	Physician, Nurse, Nurse Aides, Occupational Therapist, Physiotherapist (n = 172)	Psychologists, Social Workers, Fear of contagion; Higher workload and high staff and Personal Protective Equipment (PPE)
Okoli and Seng [69]	Post-traumatic Growth Inventory [9]	Advanced practice providers; Nursing staff; Social work/ Psychology; Nursing care technicians/nursing assistants; Therapists (occupational/recreational/physical/ respiratory, paramedics, technicians); Pharmacy; Nondirect care staff (e.g., clerical staff or administration) (n= 479)	Demographic variables (Gender (female); nonheterosexual; Older age; post-graduate degree; having Children). Work related variables (nondirect care employees; working in pediatric care). Behavioral variables (lower alcohol consumption; having had treatment for trauma)
Okoli et al. [70]	Post-traumatic Growth Inventory [9]	Nurses (n = 299)	Having a postgrad degree; serving the pediatric population; lower frequency of alcohol use.
Peng et al. [71]	Post-traumatic Growth Inventory [9]	Nurses (n = 116)	Having children, physical discomfort and getting support from family and friends during the epidemic.
Plews-Ogan et al. [72]	Grounded theory analysis of interviews	Physicians (n = 61)	Sharing (talking about) experience; Wisdom
Shiri, Wexler and Kreitler, [73]	Post-traumatic Growth Inventory [9]	Rescuers, nurses and rehabilitation workers (n = 51)	Beliefs rooted in optimism
Simmons et al. [74]	Post-traumatic Growth Inventory [9]	Military nurses (n = 119)	Spirituality
Taku [75]	Short form of the PTG Inventory [52]	Physicians (n = 289)	Personal accomplishment
Xu et al. [76]	Post-traumatic Growth Inventory [9]	Physicians, nurses and other medical workers (n = 579)	Receiving psychological intervention or training: Positive coping strategies for emotion management (i.e., exercising, paying attention to the positive aspects of the event and talking with friends); Cultivating optimism; Meditation and muscle relaxation techniques to manage stress; Sharing growth with others.
Yılmaz, Üstün and Günüşen [77]	Post-traumatic Growth Inventory [9]	Nurses (n = 43)	Receiving psychological intervention or training Meaningful self-reflection and fostering self-awareness (cultivated through educational lecture exercises, baksi dance and using mandala painting techniques)
Zhang et al. [78]	Post-traumatic Growth Inventory [9]	Nurses (1790)	Social support; Self-efficacy

## Data Availability

Not applicable.

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
