# Peer review of "Factors Associated with Post-Traumatic Growth in Healthcare Professionals: A Systematic Review of the Literature"

_healthcare, 2022, doi:10.3390/healthcare10122524_

Round 1
Reviewer 1 Report (Previous Reviewer 2)
The authors used adequate methodology and included enough details in different sections with summary tables.
Author Response
Dear Reviewer,
Thank you for the careful reading and considered review of our manuscript. We are very grateful for their time and valuable suggestions for improvements. Our revised manuscript has been reviewed to address any English language, style or spelling errors.
Reviewer 2 Report (New Reviewer)
Posttraumatic growth (PTG) research is an emerging healthcare issue as recent COVID-19 has resulted in some trauma to healthcare professionals. This study systematically reviewed 27 papers out of 126 published works. Notably, they considered a plethora of factors to take into consideration in their meta-analysis. This meta-analysis and interpretation of the data are also very coherent. This review work can provide a guideline for coping with PTG among healthcare professionals. Though this study has merit, the authors seem rushed to submit the manuscript without substantial editing.
Specific comments:
- The authors performed this study that answered the research question.
- Overall, this paper needs substantial editing of grammar and language before being published. A few examples,
• In line 44, “The conceptualisations of posttraumatic growth range from experiencing stress-related growth” the spelling of conceptualization is wrong.
- In line 61, “However, the measure of PTG Inventory has been since used to assess secondary (SPTG) or vicarious posttraumatic growth (VPTG), denoting positive changes experienced following on from observing other people experiencing trauma, which is often the case with healthcare professionals” here the word “since used” appears to be grammatically incorrect.
- The authors mentioned that in their meta-analysis of 26 studies, 10-77.3% of participants experienced PTG. It could have been better to have a possible explanation of this dramatic variation of PTG among different studies and what are the possible causes of this variation.
- Importantly, this study incorporated only three qualitative components, limiting our understanding of the relationship between PTG and the enablers identified thus far.
Author Response
Dear reviewer,
Thank you for the careful reading and considered review of our manuscript. We are very grateful for their time and valuable suggestions for improvements. Please see our response to your specific comments below.
- The authors performed this study that answered the research question.
Thank you.
- Overall, this paper needs substantial editing of grammar and language before being published. A few examples,
- In line 44, “The conceptualisations of posttraumatic growth range from experiencing stress-related growth” the spelling of conceptualization is wrong.
- In line 61, “However, the measure of PTG Inventory has been since used to assess secondary (SPTG) or vicarious posttraumatic growth (VPTG), denoting positive changes experienced following on from observing other people experiencing trauma, which is often the case with healthcare professionals” here the word “since used” appears to be grammatically incorrect.
Thank you for highlighting these issues. We have revised and edited the full manuscript, along with addressing the examples you raised here.
- The authors mentioned that in their meta-analysis of 26 studies, 10-77.3% of participants experienced PTG. It could have been better to have a possible explanation of this dramatic variation of PTG among different studies and what are the possible causes of this variation.
This comment appears in our introduction section and refers to a previously conducted meta-analysis which summarised the prevalence of moderate to high PTG in people who experienced traumatic events. It is not a result found in the systematic review that we have conducted. However, we have added a comment on the possible explanation for the variation found in that previous study (line 71).
- Importantly, this study incorporated only three qualitative components, limiting our understanding of the relationship between PTG and the enablers identified thus far.
Yes, we have updated our discussion to highlight and further expand on the need for further qualitative research in this area (lines:435-444).
Reviewer 3 Report (New Reviewer)
Dear Editors,
Have a nice day.
The current literature review focused on identifying 113 factors that facilitate post-traumatic growth (PTG) among healthcare professionals. The study highlights vulnerable healthcare professional groups that require additional support in helping them experience PTG. This is a nicely crafted systematic review that meets the technicalities involved in such a work. I suggest minor adjustments.
Title: the title provided for this work is representative.
Abstract: the abstract can be expanded a bit by providing some description of the results part.
Introduction: the introduction provides a sufficient background for the study.
Methods: Figure 1 can be part of the materials and methods part rather than providing results.
Results: Table 1 uses two different styles of reference. From 'Moreno-Jiménez et al. (2021)' onward formatting changed.
Conclusion: it concludes what it sets about in the introduction.
Author Response
Dear reviewer,
Thank you for the careful reading and considered review of our manuscript. We are very grateful for their time and valuable suggestions for improvements. Please see our response to your specific comments below.
Title: the title provided for this work is representative.
Thank you.
Abstract: the abstract can be expanded a bit by providing some description of the results part.
We have now provided more details from our results in the abstact.
Introduction: the introduction provides a sufficient background for the study.
Thank you.
Methods: Figure 1 can be part of the materials and methods part rather than providing results.
We have now moved Figure 1 to the material and methods section.
Results: Table 1 uses two different styles of reference. From 'Moreno-Jiménez et al. (2021)' onward formatting changed.
Thank you for highlighting this, we have now updated all references.
Conclusion: it concludes what it sets about in the introduction.
Thank you.
Reviewer 4 Report (New Reviewer)
This is pertinent and necessary work. The nuance that it introduces in the research, identifying the factors that predict their experiences of Factors Associated with Post-Traumatic Growth (PTG), gives the work a perspective that requires its replication in other fields of health, especially in mental health.
Author Response
Dear reviewer,
Thank you for the careful reading and considered review of our manuscript. We are very grateful for their time and valuable comments.
Reviewer 5 Report (New Reviewer)
Thank you for the opportunity to read this paper, which presents a systematic review of the literature on post-traumatic growth among healthcare professionals. This is a timely topic with relevance to health service administration and medical training. The review here has been conducted well; however some more work needs to be done around it, in justifying its importance and approach taken, and in grounding findings more broadly in key phenomena of healthcare work.
Introduction: Good definition and establishment of PTG through literature, with critical consideration. However, the grounds for examining PTG, and doing the review, are not made strongly enough. Partly this comes down to the way generalisations have been made from the start, “All” HCPs, it is claimed, suffered trauma, and “35%” suffered PTSD.It might be fairer to say “many” rather than “all” HCPs experienced trauma in COVID (line 19), and rather than simply “35%” (line 21; this expression suggests that this is 35% of all HCPS everywhere) specify “in excess of 35% of some recent studies [1,2]”. One study of PTG among Hong-Kong-based HCPs after SARS suggests positive psychological development out of direct experience of contracting the virus is introduced as the foundation (without any consideration of the cultural philosophical context of this, or any clear-cut equivalence between having SARS and the “trauma” of working through COVID) but the justification between these is weak.
Methods: Some sections need stronger justification. Why was the MMAT used (line 155), why is it the most suitable and appropriate for this study? Why use a narrative synthesis than a thematic synthesis in this case? (Particularly as reference is repeatedly made to “themes” here). Do the two reviewers have expertise in psychology, in health services research, etc?. What languages were the included studies limited to, if any?
Findings: Diagrammes clear and helpful. The written description of some of the findings is disjointed (possibly due to the 4-level subheading style used which goes down possibly too finely) when the short text descriptions of each category are too abruptly cut off.
Discussion: The importance of the review is well established, particularly the way it has identified a clear differentiation between personal factors and work-related factors. Often in high-stress healthcare roles, personal issues like “resilience” and “burnout” become the preferred organisational way of thinking about stress/trauma rather than systems issues which sit behind them, and burnout happens in healthcare despite high resilience rather than because of a lack of resilience (see https://jamanetwork.com/journals/jamanetworkopen/fullarticle/2767829 by West et al). More could be made of this fact and the contribution this review can make to supporting healthcare workers and their psychological safety and growth.
One consideration that the discussion should made, and has not addressed: what role do the professional cultures of medicine/healthcare play in this? If PTG is about “thriving” rather than just “surviving” after trauma, do HCPs work in the kinds of professional culture that value “thriving”? Or is anxiety/stress/trauma normalised and, in some cases, lauded as success within these professional cultures? (See Beneath the White Coat: Doctors, Their Minds and Mental Health, edited by Clare Gerada, and https://www.mayoclinicproceedings.org/article/S0025-6196(19)30345-3/fulltext by Shanafelt for insights into this from medicine specifically)
The call for additional qualitative study (line 429-30) is welcome, though a stronger justification for why this kind of research would bring additional depth to understanding in this area could be made. Consider, if formatting will allow, a clear text box or sub-heading with direct implications/recommendations for practice and policy, to maximise the potential impact of the findings.
General: Ensure consistency – either COVID-19 or Covid-19 (but not interchangeably, just stick with one)
Recommend a thorough proof-read, as the expression and punctuation of the piece throughout has frequent errors, and is often awkward and could flow more smoothly.
Author Response
Dear reviewer,
Thank you for the careful reading and considered review of our manuscript. We are very grateful for their time and valuable suggestions for improvements. Please see our response to your specific comments below.
Introduction: Good definition and establishment of PTG through literature, with critical consideration. However, the grounds for examining PTG, and doing the review, are not made strongly enough. Partly this comes down to the way generalisations have been made from the start, “All” HCPs, it is claimed, suffered trauma, and “35%” suffered PTSD.It might be fairer to say “many” rather than “all” HCPs experienced trauma in COVID (line 19), and rather than simply “35%” (line 21; this expression suggests that this is 35% of all HCPS everywhere) specify “in excess of 35% of some recent studies [1,2]”. One study of PTG among Hong-Kong-based HCPs after SARS suggests positive psychological development out of direct experience of contracting the virus is introduced as the foundation (without any consideration of the cultural philosophical context of this, or any clear-cut equivalence between having SARS and the “trauma” of working through COVID) but the justification between these is weak.
We have rewritten our introduction to remove these generalisations at the start. We have removed our reference to direct experience of SARS among Hong-Kong based HCPs as we understand that it disctracts from the primary focus of our review. Thank you for highlighting this issue.
Methods: Some sections need stronger justification. Why was the MMAT used (line 155), why is it the most suitable and appropriate for this study? Why use a narrative synthesis than a thematic synthesis in this case? (Particularly as reference is repeatedly made to “themes” here). Do the two reviewers have expertise in psychology, in health services research, etc?. What languages were the included studies limited to, if any?
Thank you, we have now addressed each of these points in our methodology section.
Findings: Diagrammes clear and helpful. The written description of some of the findings is disjointed (possibly due to the 4-level subheading style used which goes down possibly too finely) when the short text descriptions of each category are too abruptly cut off.
We have rewritten the results sections to remove the 4-level sub-heading style. We hope it reads more clearly now.
Discussion: The importance of the review is well established, particularly the way it has identified a clear differentiation between personal factors and work-related factors. Often in high-stress healthcare roles, personal issues like “resilience” and “burnout” become the preferred organisational way of thinking about stress/trauma rather than systems issues which sit behind them, and burnout happens in healthcare despite high resilience rather than because of a lack of resilience (see https://jamanetwork.com/journals/jamanetworkopen/fullarticle/2767829 by West et al). More could be made of this fact and the contribution this review can make to supporting healthcare workers and their psychological safety and growth.
Thank you for this suggestion, we have now further expanded on this point in our discussion (lines:394-400).
One consideration that the discussion should made, and has not addressed: what role do the professional cultures of medicine/healthcare play in this? If PTG is about “thriving” rather than just “surviving” after trauma, do HCPs work in the kinds of professional culture that value “thriving”? Or is anxiety/stress/trauma normalised and, in some cases, lauded as success within these professional cultures? (See Beneath the White Coat: Doctors, Their Minds and Mental Health, edited by Clare Gerada, and https://www.mayoclinicproceedings.org/article/S0025-6196(19)30345-3/fulltext by Shanafelt for insights into this from medicine specifically)
Yes, this is an important point, we have added this to our discussion section.
The call for additional qualitative study (line 429-30) is welcome, though a stronger justification for why this kind of research would bring additional depth to understanding in this area could be made. Consider, if formatting will allow, a clear text box or sub-heading with direct implications/recommendations for practice and policy, to maximise the potential impact of the findings.
We have updated our discussion to highlight and further expand on the need for further qualitative research in this area (lines:435-444). We have highlighted implications for practice and policy more clearly in our discussion section.
General: Ensure consistency – either COVID-19 or Covid-19 (but not interchangeably, just stick with one)
Thank you for highlighting this inconsistency. We have updated the manuscript to use COVID-19 version throughout.
Recommend a thorough proof-read, as the expression and punctuation of the piece throughout has frequent errors, and is often awkward and could flow more smoothly.
Thank you, a full and thorough proof-read has now been completed.
Round 2
Reviewer 5 Report (New Reviewer)
Thank you to the authors for engaging so openly and thoroughly with previous feedback, especially with multiple reviewers comments to juggle. The revised paper is excellent and I have no hesitations in recommending acceptance for publication.
This manuscript is a resubmission of an earlier submission. The following is a list of the peer review reports and author responses from that submission.
Round 1
Reviewer 1 Report
In the present systematic review, O’Donovan and Burke have analyzed the factors associated with posttraumatic growth (PTG) in healthcare professionals. Selecting 27 papers, they list all potential factors that could be associated to PTG.
The manuscript is well written but in its current form, there are some points that are not completely clear to me:
- A number of the 27 selected papers are about healthcare professionals working during the current COVID-19 pandemic. This, the authors also mention a couple of times. However, the do not at all discuss what kind of trauma, the other papers analyze (e.g. working during war, working in an ICU and so on). I would have liked a discussion about what kind of trauma the healthcare professionals experienced and how this could affect the results.
- Or did the authors specifically want to focus on trauma during the COVID-19 pandemic (first sentence in the introduction is about COVID-19)? However, then it doesn’t make sense with other papers that are not about working during the COVID-19 pandemic.
- The 27 selected papers seem to be very different and therefore difficult to compare. Instead, the authors just list all factors mentioned in the individual papers without much discussion. For example, what does ‘wisdom’ mean and how is this different from ‘age’ and ‘years of working’? Maybe, this could be discussed further?
Minor pints:
Lines 10/11: Now, it sounds as if only 27 papers are cited in this review. However, as this is not the case, this sentence should be rephrased.
Line 30: SARS-1 was an epidemic but not a pandemic.
I suggest to move the section ‘Posttraumatic growth……. adverse situation”, lines 37-43 to the beginning of the introduction.
For some of the refs in Table 1, the reference number is missing, e.g. Moreno-Jimenez et al, 2021.
Line 248: Which pandemic is meant, the COVID-19 pandemic?
Author Response
We would like to thank you for the careful reading and considered review of our manuscript. We are very grateful for your time and valuable suggestions for improvements. Please find below our response and an account of the changes that have been made to the manuscript based on the comments received.
The manuscript is well written but in its current form, there are some points that are not completely clear to me:
A number of the 27 selected papers are about healthcare professionals working during the current COVID-19 pandemic. This, the authors also mention a couple of times. However, the do not at all discuss what kind of trauma, the other papers analyze (e.g. working during war, working in an ICU and so on). I would have liked a discussion about what kind of trauma the healthcare professionals experienced and how this could affect the results.
Or did the authors specifically want to focus on trauma during the COVID-19 pandemic (first sentence in the introduction is about COVID-19)? However, then it doesn’t make sense with other papers that are not about working during the COVID-19 pandemic.
Thank you for highlighting this important point for discussion. Our aim was not to focus on any specific type or source of trauma (i.e. COVID-19 pandemic). We wished to focus on healthcare professionals experience of PTG following any type of trauma, thus there are various types of trauma included in our review. Moreover, research on PTG rarely considers specific trauma-type, given that trauma is a subjective, not objective experience, meaning that when two people experience an event, for one of them, it may be perceived as traumatic, whereas the other person might see it as a mere hassle. Also, researchers warn about differentiating between trauma types given that when focusing on one type of traumatic experience (e.g. Covid) we may not consider personal trauma that individuals deal with. Based on all these reasons, we are not differentiating between various types of trauma. However, we have added sentences in the literature review and discussion to clarify that the paper delves into other than Covid-related trauma as well.
The 27 selected papers seem to be very different and therefore difficult to compare. Instead, the authors just list all factors mentioned in the individual papers without much discussion. For example, what does ‘wisdom’ mean and how is this different from ‘age’ and ‘years of working’? Maybe, this could be discussed further?
Thank you, we have now added further discussion of the identified enablers to the beginning of our discussion section.
Lines 10/11: Now, it sounds as if only 27 papers are cited in this review. However, as this is not the case, this sentence should be rephrased.
This sentence has now been rephrased to explain that 27 papers met the inclusion criteria for the systematic review.
Line 30: SARS-1 was an epidemic but not a pandemic.
Thank you, this has now been corrected.
I suggest to move the section ‘Posttraumatic growth……. adverse situation”, lines 37-43 to the beginning of the introduction.
We have now moved this paragraph to the beginning of the introduction.
For some of the refs in Table 1, the reference number is missing, e.g. Moreno-Jimenez et al, 2021.
These references have now been added.
Line 248: Which pandemic is meant, the COVID-19 pandemic?
Yes, this has now been clarified.
We hope the revised manuscript satisfactorily addresses your comments and look forward to hearing from you in due course.
Yours sincerely,
The authors.
Reviewer 2 Report
Overall, it's well written. The authors used adequate methodology and included enough details in different sections with summary tables and figures. Below is one comment for the authors to consider.
This meta-analysis is based on PTG in the original context as Tedeshchi and Calhoun’s model (line 60). It’s not addressing secondary (SPTG) or vicarious posttraumatic growth (VPTG). According to results there are 19 cross sectional, 3 longitudinal studies. Among those, there are six studies met the quality criteria. I am wondering whether it’s possible to generate a forest plot for these six studies to compare common factors related to PTG measure. If required information not available or if it is not possible then it’s okay.
Author Response
We would like to thank you for the careful reading and considered review of our manuscript. We are very grateful for your time and valuable suggestions for improvements. Please find below our response and an account of the changes that have been made to the manuscript based on the comments received.
Overall, it's well written. The authors used adequate methodology and included enough details in different sections with summary tables and figures. Below is one comment for the authors to consider.
This meta-analysis is based on PTG in the original context as Tedeshchi and Calhoun’s model (line 60). It’s not addressing secondary (SPTG) or vicarious posttraumatic growth (VPTG).
Yes, we have now clarified in the methodology section that SPTG and VPTG were beyond the scope of the current systematic review. We also wish to clarify that this is a systematic review, no a meta-analysis.
According to results there are 19 cross sectional, 3 longitudinal studies. Among those, there are six studies met the quality criteria. I am wondering whether it’s possible to generate a forest plot for these six studies to compare common factors related to PTG measure. If required information not available or if it is not possible then it’s okay.
Thank you for this suggestion. However, given the nature of this article, this would be neither possible nor appropriate. We have conducted a systematic review of the literature, not a meta-analysis. Additionally, given the heterogeneity in the studies included in this review, a narrative approach to analysis was deemed most appropriate. We believe that it would not be appropriate or accurate to combine the analysis of the wide variety of studies included in this review in the form of a meta-analysis.
We hope the revised manuscript satisfactorily addresses your comments and look forward to hearing from you in due course.
Yours sincerely,
The authors.
Reviewer 3 Report
This systematic review of the PTG in healthcare professionals is a very interesting approach to gain a general (and so far lacking) overview of this concept and its importance in healthcare.
The used methodology is interesting and rigorous. The main revision should concern the discussion of the results, which in its current form, is insufficient.
Introduction
« PTG occurs when adverse events are central to individuals’ self-identity »: Could you be more specific? This idea should be developed further.
« Three main types of intervention approach for enhancing PTG are (1) self-expression or disclosure – written or spoken, (2) cognitive behavioural therapy, and (3) novel psychological therapies aimed to facilitate PTG »: Please be more specific: In which population? Patients? Healthcare professionals?
In the literature review the results are sometimes confusing, as the authors don’t differentiate between trauma patients and healthcare professionals. Because of professionals’ coping strategies and other professional specific resources, we can expect that their potential PTG is experienced differently than a patient's PTG.
Materials and methods
How many nurses were included in the reviewed studies in total?
Could the The Mixed Methods Appraisal Tool be described?
How many researchers were involved in the narrative approach to synthesis? Could you give more details about this process?
Results
The transition between Figure 1 and the description of the individual level factors (3.1) is very abrupt. Could this part be introduced by a brief and categories summarizing sentence?
Discussion:
The non-work related factors could be discussed further, especially as many of them were identified in the studies.
Some results were surprising (for example the link between physical discomfort and higher PTG scores; the link between high workload and lack of staff and higher PTG scores) and should be discussed.
In general, the identified factors are diverse, multiple and not always coherent and should be discussed in regard to the complexity of the PTG concept.
PTG was mainly discussed in the context of COVID care. Could we expect different results according to different types of trauma: interpersonal trauma, war trauma, sexual trauma etc? We can imagine that the possibility of PTG would also be influenced by the nature of the encountered trauma in patients or in healthcare situations.
Author Response
We would like to thank you for the careful reading and considered review of our manuscript. We are very grateful for your time and valuable suggestions for improvements. Please find below our response and an account of the changes that have been made to the manuscript based on the comments received.
his systematic review of the PTG in healthcare professionals is a very interesting approach to gain a general (and so far lacking) overview of this concept and its importance in healthcare.
The used methodology is interesting and rigorous. The main revision should concern the discussion of the results, which in its current form, is insufficient.
Introduction
« PTG occurs when adverse events are central to individuals’ self-identity »: Could you be more specific? This idea should be developed further.
« Three main types of intervention approach for enhancing PTG are (1) self-expression or disclosure – written or spoken, (2) cognitive behavioural therapy, and (3) novel psychological therapies aimed to facilitate PTG »: Please be more specific: In which population? Patients? Healthcare professionals?
In the literature review the results are sometimes confusing, as the authors don’t differentiate between trauma patients and healthcare professionals. Because of professionals’ coping strategies and other professional specific resources, we can expect that their potential PTG is experienced differently than a patient's PTG.
Thank you for your comment. We have clarified these points further.
Materials and methods
How many nurses were included in the reviewed studies in total?
There were a total of 18, 285 nurses or nursing assistants included, this has now been indicated in our methodology section.
Could the The Mixed Methods Appraisal Tool be described?
Yes, we have now further described this tool.
How many researchers were involved in the narrative approach to synthesis? Could you give more details about this process?
Both authors were included in this process. We have now provided further details about this process in the methodology section.
Results
The transition between Figure 1 and the description of the individual level factors (3.1) is very abrupt. Could this part be introduced by a brief and categories summarizing sentence?
Yes, we have now included this summary sentence as a transition into the description of individual level factors.
Discussion:
The non-work related factors could be discussed further, especially as many of them were identified in the studies.
We have now added further discussion on the non-work-related factors identified in this review.
Some results were surprising (for example the link between physical discomfort and higher PTG scores; the link between high workload and lack of staff and higher PTG scores) and should be discussed.
Thank you, we have included a paragraph discussing these findings.
In general, the identified factors are diverse, multiple and not always coherent and should be discussed in regard to the complexity of the PTG concept.
Thank you, we have now added a section to our discussion that addresses the variety of identified enablers and the complexity of PTG.
PTG was mainly discussed in the context of COVID care. Could we expect different results according to different types of trauma: interpersonal trauma, war trauma, sexual trauma etc? We can imagine that the possibility of PTG would also be influenced by the nature of the encountered trauma in patients or in healthcare situations.
We did not differentiate between covid-related and other forms of trauma, nor did we aim to include studies that discussed Covid. However, given the growing interest in PTG, more research emerged in the last couple of years, which is why there is more mention of it in our paper. Thank you for your feedback. Following on from it, we have entered a few sentences in the literature review and discussion to explicitly mention Covid and other types of trauma.
We hope the revised manuscript satisfactorily addresses your comments and look forward to hearing from you in due course.
Yours sincerely,
The authors.
Reviewer 4 Report
The authors have attempted to determine" Factors Associated with Posttraumatic Growth in Healthcare 2 Professionals: A Systematic Review of the Literature ".
This article is valuable because it is a systematic review, but generally does not indicates any new data. It has not novelty and I cannot recommend it for publication.
Author Response
We would like to thank you for the careful reading and considered review of our manuscript. We are very grateful for your time and valuable suggestions for improvements. Please find below our response and an account of the changes that have been made to the manuscript based on the comments received.
The authors have attempted to determine" Factors Associated with Posttraumatic Growth in Healthcare 2 Professionals: A Systematic Review of the Literature ". This article is valuable because it is a systematic review, but generally does not indicate any new data. It has not novelty and I cannot recommend it for publication.
Thank you for acknowledging the value of our systematic review. The purpose of a systematic review is no to add any new data but to synthesise findings from previous studies in order to assess the current state of art, to answer specific research questions and to identify gaps in the research to inform future studies (Liberati et al. 2009; Moher et al. 2009). Our review makes a significant contribution by synthesising the enablers of PTG among healthcare professionals that have been identified to date. Doing this has allowed us to discuss gaps in current research, implications for future research, policy and practice and implications for future interventions to facilitate the experience of PTG among healthcare professionals.
We hope the revised manuscript satisfactorily addresses your comments and look forward to hearing from you in due course.
Yours sincerely,
The authors.
References
Liberati A, Altman DG, Tetzlaff J., et al. The PRISMA statement for reporting systematic reviews and meta analyses of studies that evaluate health care interventions: explanation and elaboration. PLoS Med 2009:6(7):e1000100
Moher D, Liberati A, Tetzlaff J., et al. Prisma Group. Preferred reporting items for systematic reviews and meta-analyses: the PRISMA statement. PLoS Med 2009:21;6(7):e1000097.
Round 2
Reviewer 1 Report
The authors satisfactorily answered all my questions.
Before the manuscript can be recommended for publication, I just noticed a very small mistake in line 227, where it says "five all". I assume this should be just "all"?
Reviewer 3 Report
"However, recent studies suggest that healthcare populations use similar approaches to the other groups."
Could you please specify which studies, include the references?
"This tool allowed the authors to assess the quality of five all studies by answering the questions relevant to the methodology used by each individual study (mixed methods or quantitative descriptive)."
I don't understand this sentence. Are five or all studies concerned?
Reviewer 4 Report
The authors' response did not convince me. My opinion is still the rejection of the article.